# Quantitative Dissection of Relaxation Processes in Hybrid Epoxy Composites: Combining Dielectric Spectroscopy with Activation Energy Analysis

**DOI:** 10.3390/polym17101405

**Published:** 2025-05-20

**Authors:** Xingqiao Li, Hongliang Zhang, Yansheng Bai, Hai Jin, Hong Wang, Kangle Li, Xiaonan Li

**Affiliations:** College of Electrical Engineering and Information Engineering, Lanzhou University of Technology, Lanzhou 730000, China

**Keywords:** epoxy resin, toughening, dielectric, activation energy

## Abstract

The dielectric relaxation dynamics in polymer composites critically determine their functional performance in advanced electrical systems. This study systematically investigates hybrid epoxy composites comprising neat epoxy resin (EP) and paper-reinforced systems (EIP), modified with 10–50 wt% polypropylene glycol diglycidyl ether (PEGDGE) plasticizer. Through synergistic application of differential scanning calorimetry (DSC) and broadband dielectric spectroscopy (10^−1^–10^6^ Hz), the quantitative relationships between plasticizer content, glass transition temperature (Tg), and dielectric relaxation processes were established. DSC analysis reveals a linear Tg dependence with increasing PEGDGE content, attributed to enhanced molecular mobility. Dielectric characterization demonstrates three distinct relaxation regimes: α-relaxation below Tg, interfacial polarization at epoxy/PEGDGE boundaries, and paper/epoxy interfacial effects in EIP systems. A quantitative dielectric relaxation model was developed based on complex modulus formalism, coupled with Vogel–Fulcher–Tammann (VFT) analysis of DC conductivity. Activation energy mapping through Arrhenius decomposition reveals three characteristic values: (1) 82.01–87.80 kJ/mol for α-relaxation, (2) 55.96–64.64 kJ/mol for epoxy/PEGDGE interfaces, and (3) 30.88–44.38 kJ/mol for epoxy/paper interfaces. Crucially, the plasticizer content modulates these activation energies, demonstrating its role in tailoring interfacial dynamics.

## 1. Introduction

Ultra-High-Voltage Direct Current Transmission (UHVDC) power transmission technology is widely applied in China and has been leading in the world, with voltage level covering from ±800 kV to ±1100 kV [1,2]. As the main insulating material in the UHV converter transformer bushing, epoxy resin (EP) composites must withstand high temperature, high electric field, and continuous action of electric field force for a long time. Although EP possesses excellent mechanical properties through complex crosslinking reactions in the curing process, this highly crosslinked structure also makes the material tend to be brittle. Moreover, when the EP composites are subjected to external impact or continuous stress, it is easy to produce cracks, which may lead to the overall fracture or cracking of the material [3,4,5], thus seriously affecting the operation safety of the bushing. Therefore, good toughness of EP composites becomes a necessity. Current methods to improve the toughness of EP composites mainly include thermoplastic resin toughening [3,4], nanoparticle toughening [6,7], etc. After adding filler into EP, its relaxation process may change due to the variation in free volume and the filler affecting the local movement of chain segments, pendant groups, and main chain, while the relaxation transition of composites at all levels will affect their dielectric properties and surface charge accumulation and attenuation, thus determining the service environment, service life, and stability of composites. Hence, the relaxation behavior of toughened EP composites for UHV converter transformer bushing is of great significance for further improving the design, manufacturing, operation, maintenance, and test of UHV bushing. Previous studies mainly analyzed the mechanical and thermal properties of toughened EP composites; in recent years, research emphasis has been shifted to the dielectric properties of toughened EP composites [8,9,10,11]; Lyu et al. investigated the mechanical and dielectric properties of epoxy resins after being toughened with different types of vegetable oils and found that the soybean-oil-based thermosets had a good tensile strength (7.5 ± 0.28 MPa), a glass transition temperature (Tg) of about 23 °C, and a dielectric constant of about 2.95 at 10 MHz [12]. Wu et al. prepared six epoxy/anhydride samples and evaluated the medium-frequency transformer (MFT) of the dielectric loss of the MFT. It was found that anhydrides containing benzene rings or methyl substituents significantly reduced the dielectric loss, with the lowest loss (16.2 W) for EP cured with methylnadiene anhydride (MeNA). The addition of anhydrides, especially with a benzene ring or methyl group, significantly reduces the temperature at which the β relaxation peak appears and enhances the chain segment rigidity, thus reducing the dielectric loss [13]. Rehim et al. investigated the dielectric properties of graphene-modified epoxy resins, and it was shown that α-polarization was not observed and that the ε′ of both unmodified and modified EPs increased slightly with the temperature and that the dielectric loss spectra of low-temperature ε″(ν) shows a microcapacitor effect, which is due to the occurrence of interfacial polarization within the composite [14]. Wang et al. investigated four composites of liquid rubber/EP with different polarities and found that, the lower the polarity of the liquid rubber, the lower the relative permittivity and dielectric loss of the composites and, also, the polarity of the liquid rubber affects the α-polarization strength of the composite and the relaxation time of the epoxy/rubber interfacial polarization [15]. Existing studies have mainly focused on the dielectric properties after toughening and modifying EP with different fillers, but there are fewer studies on how the ratio of the same toughening agent affects the relaxation process and quantitatively analyzes the type of relaxation based on parameters such as the activation energy of dielectric relaxation.

Studies have shown that adding polyethylene glycol (PEG) [16] and diethylene glycol flexible diglycidyl ether (DGEG) [17] to epoxy matrices can significantly improve the impact and fracture properties of epoxy thermosetting polymers. In this work, epoxy resins and epoxy-impregnated paper containing 10%, 30%, and 50% polyethylene glycol diglycidyl ether (PEGDGE) were prepared. Unlike traditional toughening agents such as carboxyl-terminated butadiene acrylonitrile (CTBN) rubber, which rely on petroleum-based feedstocks, the synthesis of PEGDGE can be combined with bio-based ethylene glycol, reducing fossil resource consumption. Additionally, the synthesis process of PEGDGE generates 60% lower wastewater COD values compared to conventional toughening agents (e.g., anhydride curing agents), and sodium hydroxide can be recycled in the closed-loop reaction, minimizing waste emissions. Differential scanning calorimetry (DSC) and an integrated insulation electrical testing system were employed to measure the glass transition temperature (Tg) and broadband dielectric spectra of the epoxy composites. The tensile strength of different samples was tested and the elongation at break was calculated. The real and imaginary parts of the complex dielectric constant and DC conductivity were analyzed using appropriate fitting formulas to derive relevant dielectric parameters and identify polarization mechanisms within the samples. Based on experimental and fitting results, the effects of toughener content and (Tg) on the internal polarization processes of epoxy composites were investigated and the underlying microscopic mechanisms were elucidated.

## 2. Experiment

### 2.1. Sample Preparation

In this study, E51 bisphenol-A epoxy resin from Shanghai Aotuntong Chemical Technology Co., Ltd. (Shanghai, China) was selected as the epoxy matrix. This resin has an epoxy value of 184–200 g/eq and a viscosity of 7000–18,000 Pa·s at 25 °C. Methylhexahydrophthalic anhydride (MeH-HPA) was used as the curing agent and *N*,*N*-dimethylbenzylamine (a colorless to pale yellow liquid with a low melting point of −75 °C and a relative density of 0.90 g/cm^3^) served as the accelerator. The toughening agent, polyethylene glycol diglycidyl ether (PEGDGE), with an average molecular weight of 400 g/mol, was supplied by Shanghai Macklin Biochemical Co., Ltd. (Shanghai, China). Type 67/100 crepe insulation paper from Weidmann (Rapperswil-Jona, Switzerland) was used as the substrate for epoxy-impregnated paper samples. Figure 1 shows the structural formula of the chemical reagents required for specimen preparation.

The PEGDGE contains ether bonds, which participate in the epoxy curing reaction via its terminal epoxy groups, forming strong interactions with the resin matrix. The flexible long aliphatic chains in its molecular structure enable free rotation and elasticity, effectively mitigating brittle cracking in the cured epoxy products.

Mixing: E51 epoxy resin, MeH-HPA curing agent, PEGDGE toughener, and accelerator were homogenized using a magnetic stirrer for approximately 20 min. The mass ratios of the components were 100:85:10:0.3, 100:85:30:0.3, and 100:85:50:0.3.

Degassing: The mixture was placed in a vacuum oven at 55 °C for 6 h to remove air bubbles.

Molding: A stainless-steel mold (capable of producing 10 samples per batch) was adjusted using spacers to control sample thickness. The degassed mixture was poured into the mold and cured in a vacuum oven under the following temperature profile: 85 °C (4 h) → 110 °C (4 h) → 145 °C (5 h) → 120 °C (4 h), followed by natural cooling to room temperature. The final epoxy resin samples measured 110 mm × 110 mm × 0.5 mm.

Preparation of Epoxy-Impregnated Paper Samples: To eliminate moisture interference, the crepe paper was cut to size and vacuum-dried at 105 °C for 12 h prior to impregnation. The dried paper was placed between spacers in the mold and the thickness was adjusted as required. The same resin formulation and curing protocol were applied to prepare epoxy-impregnated paper samples with identical dimensions. Figure 2 shows the process of preparing epoxy composite specimens.

### 2.2. Glass Transition Temperature

Glass transition temperature can directly reflect the difficulty level of the glass transition process of polymers. To explore the influence of PEGDGE content on the toughness of EP and EIP samples, DSC 1 of Mettler Toledo Company (Tiel, The Netherlands) was used as the test instrument to measure the Tg of EP and EIP with different contents of PEGDGE by the DSC method; the test temperature range is 30 °C~200 °C, with a temperature increase rate of 10 K/min.

### 2.3. Dielectric Test

The cured EP composites were processed into circular sheet samples with a thickness of 0.2 mm and a diameter of 50 mm, and metal electrode with a diameter of about 30 mm was uniformly sputtered on both sides of the samples by a magnetron sputtering apparatus under high vacuum conditions (10^−4^ Pa).

The integrated electrical insulation test system of Beijing HuaCe Technology Co., Ltd. (Beijing, China) was adopted as the dielectric property measurement system, which is mainly composed of Huace6630 impedance analyzer, HEST-300 high-impedance meter, high–low temperature sample table, and temperature controller. In this study, the measurement frequency range is 10^−1^ Hz~10^6^ Hz and the effective voltage value is 200 V.

### 2.4. Mechanical Performance Testing

The mechanical tests were conducted in accordance with criteria in reference [18]. An MTS E44.04 microcomputer-controlled electronic universal testing machine (MTS System, Eden Prairie, MN, USA) was employed to perform tensile testing on specimens at room temperature. The specimen dimensions were selected as Type 1A specified in the standard method, with a crosshead speed of 1 mm/min.

## 3. Test Results

Figure 3 shows the test results of glass transition temperature of epoxy composites. The Tg of EP samples with PEGDGE content of 10%, 30%, and 50% is 102 °C, 59.8 °C, and 48.8 °C, respectively, while that of EIP samples is 101.5 °C, 53.5 °C, and 40.2 °C, respectively. With the increase in PEGDGE content, the Tg of EP composites gradually decreases. Tg represents the lowest temperature of molecular segment motion. The lower the Tg, the smaller the rigidity of polymeric segment and the stronger the ability of segmental motion. This is because the crosslinking reaction of EP composites is hindered due to the addition of PEGDGE, which reduces the degree of curing, resulting in the reduced compactness and enhanced molecular activity of the cured product with a three-dimensional network structure, thereby reducing Tg. When the PEGDGE content exceeds 30%, Tg does not decrease much, because certain PEGDGE content will limit the motion of molecular chain segments.

From Table 1, it can be observed that the maximum tensile load and tensile strength decrease with increasing PEGDGE content, while the elongation at break increases significantly with higher PEGDGE content. Notably, when the content rises from 30% to 50%, the elongation at break increases by 228%, indicating that the ductility and toughness of the epoxy resin improve with higher PEGDGE content.

When the PEGDGE content is too low (10%), the epoxy resin fails to form an effective flexible network or phase-separated structure and its crosslink density remains high, restricting the plastic deformation capability of molecular chains [19]. In contrast, at a higher PEGDGE content (50%), PEGDGE forms a continuous flexible phase, absorbing energy through the “bridging-crack pinning effect” to delay crack propagation. The increased PEGDGE content may also promote the rearrangement of the epoxy resin’s crosslinked network, creating a looser “rigid-flexible transition interface” that enhances molecular chain mobility [20].

Figure 4 and Figure 5 illustrate the relationship between the real part ε′ (relative dielectric constant) and imaginary part ε″ (dielectric loss) of complex dielectric constant of EP and EIP sheets with different PEGDGE contents and frequency and temperature. The test temperature started from 20 °C and increased in a gradient style at 20 °C intervals. The dielectric properties of 10%, 30%, and 50% PEGDGE samples at 20 °C, 40 °C, 60 °C, 80 °C, 100 °C, 120 °C, and 140 °C were tested, respectively. Since the moisture in the samples has a great influence on the results of the dielectric test, the samples with sputtered electrodes were placed in a vacuum drying oven at 105 °C and dried under vacuum conditions for 12 h before the test to eliminate the moisture to the greatest extent.

The relationship between complex dielectric constant and frequency can describe the relaxation process of the molecular chain segments of EP. The complex dielectric constant is often described as the Havriliak–Negami (HN) model:(1)$$\varepsilon^{*}(f)={\varepsilon^{\prime }}_{\mathrm{HN}}\left(f\right)-i{\varepsilon^{\prime \prime }}_{\mathrm{HN}}(f)=\varepsilon_{\infty }+\frac{\Delta \varepsilon }{{\left(1+{\left(i2\pi f\tau_{\mathrm{HN}}\right)}^{\beta }\right)}^{\gamma }}$$
where $\varepsilon_{HN}^{\prime }$ and $\varepsilon_{HN}^{\prime \prime }$ represent the real part and imaginary part of the complex dielectric constant in the HN model; *τ_HN_* denotes the dielectric relaxation time constant of the HN model; and *β* and *γ* are the shape factors characterizing the relaxation peak broadening and asymmetry effect, respectively, 0 < *β* ≦ 1, 0 < *βγ* ≦ 1.

It can be seen from Figure 4 that, under a fixed temperature, the *ε*′ and *ε*″ values of the three groups of samples gradually increase with the decrease in frequency and, the higher the temperature, the greater the increase. In Reference [21], the epoxy resin with 6% highly branched silicon hydride (QSiH) exhibits a dielectric constant of 3.2, whereas the epoxy resin with 10% PEGDGE under the same conditions shows a higher dielectric constant of 3.64. This discrepancy arises because the hydrophobic nature of organosilicon suppresses polarizability, while the ether bonds in PEGDGE contribute to enhanced dielectric polarization. Reference [22] synthesized a novel flexible chain-stopped hyperbranched polyester (HBP). When HBP content is 10% and 25%, the dielectric constants of the epoxy resin are 3.8 and 5.5, respectively. In contrast, under identical conditions, the dielectric constants for 10% and 30% PEGDGE-modified epoxy are 3.64 and 3.82. The limited compatibility between HBP and epoxy resin leads to the formation of microphase-separated structures, where interfacial charge accumulation (Maxwell–Wagner effect) significantly enhances interfacial polarization [23], thereby resulting in a higher dielectric constant.

When the temperature reaches a certain value, the *ε*″ slope of the three groups of samples is about −1 in the test frequency range, indicating that, at this temperature, the DC conductance process occurs in the frequency range; under the same testing temperature, relaxation peaks can also be observed in the *ε*″ of the samples with 10% and 30% PEGDGE, and these peaks shift toward higher frequencies as the temperature increases.

Like EP, the *ε*′ and *ε*″ values of three types of EIP samples also increase with the increase in frequency. When the temperature is greater than 100 °C, in the frequency range corresponding to the ascending section in the step-zone of *ε′*, the ε″ of samples with 10% and 30% PEGDGE also show a flat zone, indicating that these two types of samples have two relaxation processes. When the temperature is greater than 80 °C, DC conductance process appears in 50% samples. When the temperatures are 20 °C, 40 °C, and 60 °C, *ε*″ shows an obvious relaxation, indicating that 50% samples have a dielectric relaxation process in the low-frequency region at these three temperatures.

To more clearly observe the relaxation process of EP composites, the test results of the complex dielectric constant are converted into the form of complex electric modulus. Figure 6 presents the frequency spectra of the imaginary part $M^{\prime \prime }$ of the complex electric modulus.

According to the test results, when the content of PEGDGE is constant, the higher the test temperature, especially when it exceeds the glass transition temperature of the samples, the easier the relaxation peak appears in EP. When the test temperature is constant, the higher the PEGDGE content, the more likely the relaxation process will occur and the more obvious the relaxation peak moving towards the high-frequency direction with the increase in temperature, while the temperature at which DC conductance appears also decreases with the increase in PEGDGE content. It is worth noting that, when the temperature exceeds 100 °C, a less obvious relaxation peak appears in the high-frequency region of samples with 10% PEGDGE content and in the low-frequency region of samples with 30% PEGDGE, but only one relaxation peak can be observed in 50% samples in the test frequency range. It can be seen from the test results that 50% samples undergo DC conductance process at almost all test temperatures, so it is speculated that the second relaxation process is concealed by conductance loss.

According to the test results of EIP, similar to EP, the higher the test temperature of the same sample, the easier the relaxation peak appears and, the higher the content of PEGDGE, the easier the dielectric relaxation and DC conductance process occurs. Different from EP, when the PEGDGE content reaches 50%, obvious relaxation peaks can still be observed in the low-frequency region and move towards the high-frequency region with the rise in temperature.

To sum up, the higher the PEGDGE content, the larger the *ε*′ value, suggesting a higher relaxation polarization degree of the sample, and the greater the *ε*″ value, indicating more loss of the sample. This is because the curing process of epoxy resin involves a ring-opening polymerization reaction between epoxy groups and curing agents, forming a three-dimensional crosslinked network structure. The long-chain structure of PEGDGE occupies the inter-crosslink spacing, reducing the crosslinking density and reconstructing the topology of the epoxy network. The decreased crosslinking density enhances the mobility of molecular chains, expands the movement range of flexible chain segments, and increases energy dissipation, thereby leading to higher dielectric loss [24]. Judging from the glass transition temperature of the samples, the temperature at which the relaxation peaks of the EP samples with 10%, 30%, and 50% PEGDGE appear is 120 °C, 60 °C, and 40 °C, respectively, while the temperature at which the relaxation peaks appear in EIP samples is 120 °C, 80 °C, and 60 °C, and the obvious relaxation processes of the six types of samples all appear close to or higher than their glass transition temperature. When the polymer changes from a hard and brittle glass state to highly elastic rubber state, *α* relaxation is the main transition. Therefore, the obvious relaxation process of the samples under the test conditions is *α* relaxation.

## 4. Fitting Analysis

It can be seen from the test results that the EP samples containing 10% and 30% PEGDGE have two dielectric relaxation processes at high temperatures, while the samples containing 50% PEGDGE can only observe one relaxation process at all temperatures. To fit and analyze the dielectric properties of the samples, two complex dielectric constant expressions are constructed to describe the relaxation process inside the samples.

Equation (2) is a complex dielectric constant expression including one HN model and considering the influence of conductance, and Equation (3) is the setting range of each parameter in Equation (2).(2)$$\varepsilon_{\mathrm{HN}}^{*}(f)=\varepsilon_{\infty }+\frac{\Delta \varepsilon_{1}}{{\left(1+{\left(i2\pi f\tau_{\mathrm{HN}1}\right)}^{\beta_{1}}\right)}^{\gamma_{1}}}-ia(\frac{\sigma_{0}}{\varepsilon_{0}f^{s}})$$(3)$$\left\{\begin{array}{l} \varepsilon_{\infty }=\varepsilon^{\prime }{|}_{f=\max}\\ \Delta \varepsilon_{1}>0\\ \tau_{\mathrm{HN}1}>0\\ 0<\beta_{1}\leq 1\\ 0<\gamma_{1}\leq 1\\ a>0\\ 0<\sigma_{0}\leq \sigma^{\prime }{|}_{f=\min}\\ 0<s\leq 1 \end{array}\right.$$
where the value of *ε_∞_* is the real part *ε*′ value of complex dielectric constant at the highest frequency (10^6^ Hz) measured by dielectric test; Δ*ε*_1_ is the dielectric relaxation intensity of the HN model in the fitting equation; *τ*_HN1_ is the dielectric relaxation time constant of the model; *β_1_* and *γ*_1_ are the two factors mentioned in Equation (1); a is a constant; *σ*_0_ is DC conductivity and its value is the measured value of the complex AC conductivity of EP at the lowest frequency (10^−1^ Hz) at different test temperatures; and *s* is the empirical coefficient that can characterize the conductance type.

Equation (4) is a complex dielectric constant expression containing two relaxation processes, in which the HN1 and HN2 models are used to fit the dielectric relaxation process in the high-frequency region and in the low-frequency region, respectively. According to the test results of complex electric modulus, Equation (2) is selected for fitting the samples with only one relaxation process and Equation (4) for fitting the samples with two relaxation processes. Both equations use a differential evolution algorithm. Furthermore, since the fitting principle of the two equations are basically the same, to accelerate the fitting speed, the setting range of parameters in Equation (3) is the same as that in Equation (4).(4)$$\varepsilon_{\mathrm{HN}}^{*}(f)=\varepsilon_{\infty }+\frac{\Delta \varepsilon_{1}}{{\left(1+{\left(i2\pi f\tau_{\mathrm{HN}1}\right)}^{\beta_{1}}\right)}^{\gamma_{1}}}+\frac{\Delta \varepsilon_{2}}{{\left(1+{\left(i2\pi f\tau_{\mathrm{HN}2}\right)}^{\beta_{2}}\right)}^{\gamma_{2}}}-ia(\frac{\sigma_{0}}{\varepsilon_{0}f^{s}})$$

The results of two temperatures of 120 °C and 20 °C in the fitting curve are selected for analysis, as shown in Figure 7. 

It can be seen from Figure 8 that the measured values and fitted values of *ε*′ are basically coincident. Figure 8 and Figure 9 show that the fitting effect of *ε*″ is good in the high-frequency range and low-temperature environment but poor in the high-temperature environment and low-frequency range because the sample has another relaxation process at this time. The dielectric test results in the previous section indicate that, when the temperature is greater than 100 °C, the *ε*′ of six types of samples continues to rise with the decrease in frequency in the range of 0.1 Hz–1 Hz, while the slope of *ε*″ in the same frequency range reduces less obviously and the positions with poor fitting effect of *ε*″ all occur within 1 Hz, which conforms to the law mentioned in reference [25], so the relaxation process is electrode polarization.

Since low-frequency electrode polarization exerts a certain influence on the analysis of the relaxation process and this influence can be suppressed by the complex electric modulus characterization, the dielectric relaxation time constant is generally obtained according to the peak frequency of imaginary-part loss peak of complex electric modulus, and the peak frequency *f_Mp_* and relaxation time constant *τ_M_* are in the following relationship:(5)$$\tau_{M}\approx \frac{1}{2\pi f_{Mp}}$$

According to the measurement results of complex electric modulus, the relaxation time constant *τ_M_* corresponding to the peak frequency *f_Mp_* of imaginary-part $M^{\prime \prime }$ loss of complex electric modulus at different temperatures can be calculated according to Equation (5), and the time constants *τ_HN_*_1_ and *τ_HN_*_2_ of HN1 model and HN2 model can be obtained by fitting Equations (3) and (4). The HN1 model is an α relaxation process, and the three parameters varying with temperature are curved. It can be seen from Figure 10 that, at the same temperature, *τ_M_* is slightly smaller than the time constant *τ_HN_*_1_ obtained by the HN1 model fitting because the same relaxation process will move towards the high-frequency direction in the form of complex electric modulus characterization.

The variations in *τ_M_*, *τ_HN_*_1_, and *τ_HN_*_2_ with temperature all obey the Arrhenius relationship. By fitting the time constant according to the Arrhenius equation, the activation energy of the relaxation process can be obtained, which represents the difficulty of the relaxation process. Activation energy obtained by fitting the time constant *τ_M_* of $M^{\prime \prime }$ is 99.38 kJ/mol, 91.66 kJ/mol, and 85.87 kJ/mol; the activation energy corresponding to *τ_HN_*_1_ is 87.80 kJ/mol, 84.90 kJ/mol, and 82.01 kJ/mol, respectively, indicating that the energy required for α relaxation gradually decreases; when the temperature is constant, *τ_HN_*_1_ gradually decreases with the increase in PEGDGE content because, the lower the glass transition temperature, the easier the chain segment moves, thus shortening the relaxation time of α relaxation; the activation energy corresponding to the time constant *τ_HN_*_2_ of the HN2 model is 64.64 kJ/mol and 55.96 kJ/mol, respectively. According to previous research of our team on the broadband dielectric spectrum of EP composites without PEGDGE, the corresponding relaxation process cannot be found at the test temperature and frequency because, when the content of PEGDGE is small, the particle size and dispersity of the dispersion phase will increase and the epoxy/PEGDGE interface will be formed inside the sample, causing charge accumulation, thus leading to the formation of relaxation peaks. Hence, the HN2 model characterizes the epoxy/PEGDGE interfacial polarization process. When the content of PEGDGE increases from 10% to 30%, more epoxy/PEGDGE interfaces are formed, the interfacial polarization degree and ε′ value increase, and the activation energy required by interfacial polarization decreases.

The activation energy obtained by fitting the *τ_HN_*_1_ of EIP is 89.73 kJ/mol, 80.08 kJ/mol, and 53.07 kJ/mol, which is similar to the fitting results of EP. Since the glass transition temperature of samples with 30% PEGDGE is greatly reduced compared with that of samples with 10% PEGDGE, the activation energy required for α relaxation also significantly decreases. The activation energy corresponding to *τ_HN_*_2_ is 44.38 kJ/mol, 38.59 kJ/mol, and 30.88 kJ/mol, respectively. Compared with EP, the activation energy required for the HN2 relaxation process of EIP is lower, and 50% samples are not concealed by the conductivity loss, suggesting that the HN2 model of EIP is the epoxy/paper interfacial polarization, and this polarization moves towards the high-frequency region with the rise in temperature and completely coincides with the α relaxation peak when the temperature is 100 °C, which also explains that the relaxation reaches peak value at 100 °C. When the temperature exceeds 100 °C, the HN2 relaxation peak is separated from the HN1 relaxation peak and moves to the low-frequency region, which explains why the HN2 relaxation peak appears again at 140 °C.

In the dielectric test, if the characteristic frequency of the charges involved in the conductance inside the sample is greater than the test voltage frequency, then, when the test voltage frequency changes, the charge migration rate changes accordingly. Based on the test results of the real part *σ’* of complex AC conductivity made by our team, it is found that the value of *σ’* will not change with the frequency within a fixed test frequency range but will remain as a fixed value, which is the DC conductivity *σ_dc_* of the sample. There is an Almond–West relationship between the real part σ′ of AC conductivity and DC conductivity *σ_dc_* [26,27,28]:(6)$$\sigma^{\prime }\left(f\right)=\sigma_{dc}\left(1+{\left(\frac{f}{fc}\right)}^{s}\right)$$
where *f* is the voltage frequency of broadband dielectric test; *f_c_* is the characteristic frequency of the internal charge response in EP, which indicates that, under the action of voltage, the motion state of the charges inside the sample changes from passing through the pressurized electrode to continuously moving back and forth inside the sample; s is the power exponent of material relaxation. Generally, the value of s ranges from 0 to 1. When f is much smaller than *f_c_*, *σ_dc_* is equal to the value of *σ*′. When f is much larger than *f_c_*, *σ*′ and f satisfy the relationship of Equation (6). Because the glass transition temperature of EIP and EP containing 10% PEGDGE is too high, only 120 °C and 140 °C are higher than their glass transition temperature, so the DC conductivity *σ*_0_ obtained at the test temperature lower than the glass transition temperature of the sample was taken for Arrhenius fitting. For samples with 30% and 50% PEGDGE, according to the Almond–West model, the test data of *σ*′ of four types of samples were fitted by differential evolution algorithm using Equation (6). Based on *σ*_0_ and fitted *σ_dc_*, it is revealed that, at the same temperature, the DC conductivity of EP and EIP increases with the increase in PEGDGE content, as can be seen from Figure 11. At a temperature of 100 °C, the DC conductivities of epoxy resins containing 10%, 30%, and 50% PEGDGE are 2.07 × 10^−13^ S·cm^−1^, 1 × 10^−11^ S·cm^−1^, and 1.42 × 10^−9^ S·cm^−1^, respectively, while those of epoxy-impregnated paper with 10%, 30%, and 50% PEGDGE are 2.58 × 10^−13^ S·cm^−1^, 2.97 × 10^−11^ S·cm^−1^, and 5.31 × 10^−10^ S·cm^−1^, respectively. Significant differences in conductivity are observed between epoxy resin and epoxy-impregnated paper specimens with varying PEGDGE contents. Particularly, the DC conductivity of the samples with 50% PEGDGE content is significantly higher than that of other samples. This is because DC conductivity is affected by the number of impurity ions and their mobility [29]; the increase in PEGDGE increases not only the amount of impurity ions but also the free chain segments in the sample, thereby increasing the mobility of carriers [30], resulting in electrical conductivity increases with the increase in PEGDGE content.

To further study the relationship between the DC conductivity *σ_dc_* of the sample and the temperature, the fitted values of *σ_dc_* and the fitting curve changing with the inverse temperature 1/T are plotted in one graph. It can be seen that, when the test temperature is higher than the glass transition temperature of the sample, the variation in *σ_dc_* with temperature conforms to the empirical equation of the Vogel–Fulcher–Tammann (VFT) model [31]:(7)$$\sigma_{dc}=\sigma_{\infty }\exp\left(-\frac{DT_{0}}{T-T_{0}}\right)$$
where *T* is the absolute temperature; *σ_∞_* is the DC conductivity when *T* approaches infinity; *D* is a constant; and *T*_0_ is the Vogel temperature. Vogel temperatures *T*_0_ and glass transition temperatures Tg of PEGDGE/epoxy composition are shown in Table 2. Among them, the glass transition temperature of samples with 30% and 50% PEGDGE is about 50 K higher than the Vogel temperature obtained by fitting, which conforms to the law mentioned in reference [28]. Moreover, the DC conductivity of samples containing 10% PEGDGE below glass transition temperature suits the Arrhenius relationship. The activation energy in the DC conductance process of EP and EIP obtained by the Arrhenius equation fitting is about 0.55 eV and 0.67 eV, indicating that the DC conductance process is affected by temperature.

## 5. Conclusions

This study prepared EP and EIP samples with 10%, 30%, and 50% PEGDGE, tested the glass transition temperature and broadband dielectric spectra of six types of samples, obtained the relaxation types inside the samples according to the broadband dielectric properties of the samples at different temperatures and the fitted activation energy of each relaxation process, and analyzed the effects of temperature, PEGDGE content, DC conductivity, and other factors on the dielectric relaxation process in EP composites. The main conclusions are as follows:(1)The glass transition temperature of EP samples containing 10%, 30%, and 50% PEGDGE is 102 °C, 59.8 °C, and 48.8 °C, respectively, and that of EIP samples with the same PEGDGE content is 101.5 °C, 53.5 °C, and 40.2 °C, respectively, indicating that the addition of PEGDGE can enhance the activity of molecular chain segments, which is manifested by a substantial reduction in Tg, whereas, when the content reaches a certain level, it will limit the movement of molecular segments, which is manifested by a small reduction in Tg.(2)The dielectric relaxation of EP mainly consists of α relaxation and epoxy/PEGDGE interfacial polarization. As the PEGDGE content increases, Tg decreases and the number of epoxy/PEGDGE interfaces increases, so the activation energy fitted by the dielectric relaxation model in α relaxation (HN1 model) and epoxy/PEGDGE interfacial polarization (HN2 model) decreases. The dielectric relaxation of EIP mainly involves α relaxation and epoxy/paper interfacial polarization, and the activation energy of both relaxation processes also decreases. The epoxy/paper interfacial polarization moves towards the high-frequency region with the rise in temperature, completely coincides with the α relaxation peak when the temperature is 100 °C, and then moves to the low-frequency region when the temperature exceeds 100 °C.(3)The DC conductivity of EP and EIP containing 10% PEGDGE conforms to Arrhenius relationship, and the corresponding activation energy is 0.55 eV and 0.67 eV, respectively. The DC conductivity of samples with 30% and 50% PEGDGE conforms to the VFT model, and the Vogel temperature fitted by the VFT model is about 50 K higher than the glass transition temperature. The increase in PEGDGE simultaneously increases the number of impurity ions and free chain segments in the samples, thus increasing the carrier mobility, resulting in an increase in DC conductivity with the increase in PEGDGE content.

In forthcoming work, we will establish a finite element (FE) model of the converter transformer valve-side bushing. The dielectric characterization data of epoxy resin composites obtained in this study will be methodologically integrated into the FE framework to systematically investigate the temperature field and electric field distributions under electro-thermal coupling conditions. This integrated approach will advance our understanding of multi-physics interactions in UHVDC equipment and ultimately provide engineering references for optimized manufacturing protocols and maintenance strategies of bushings.

## Figures and Tables

**Figure 1 polymers-17-01405-f001:**
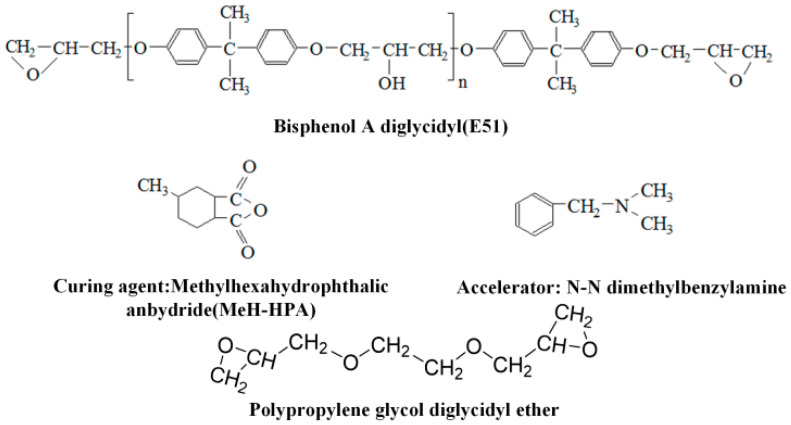
Raw materials required for sample preparation.

**Figure 2 polymers-17-01405-f002:**
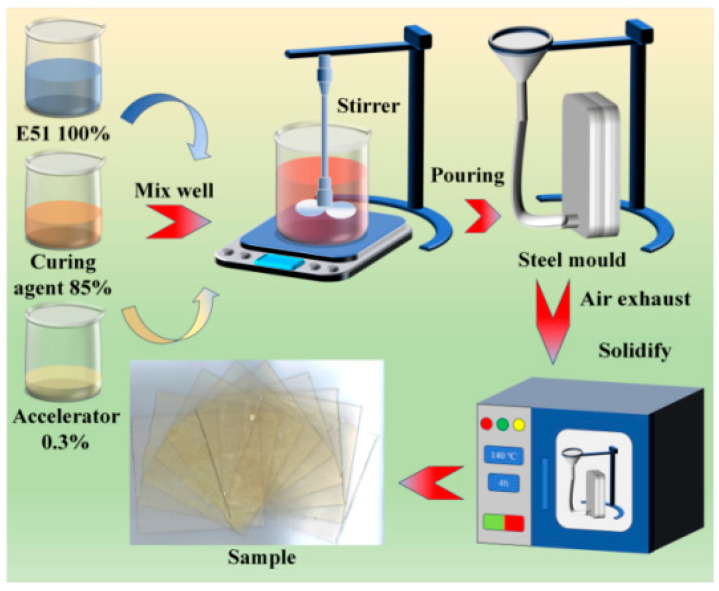
Process of sample preparation.

**Figure 3 polymers-17-01405-f003:**
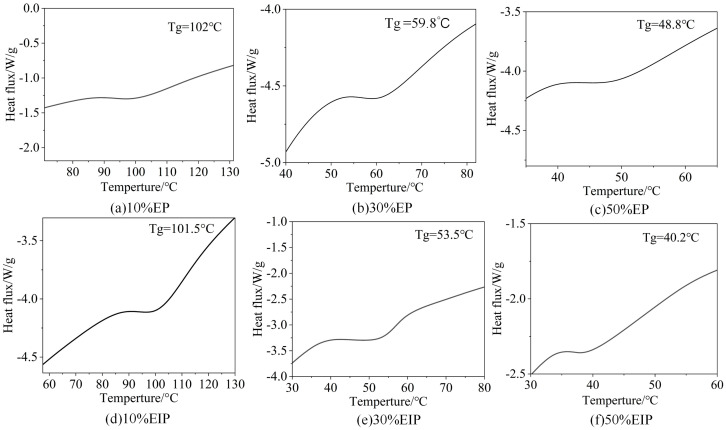
Glass transition temperature of epoxy composites.

**Figure 4 polymers-17-01405-f004:**
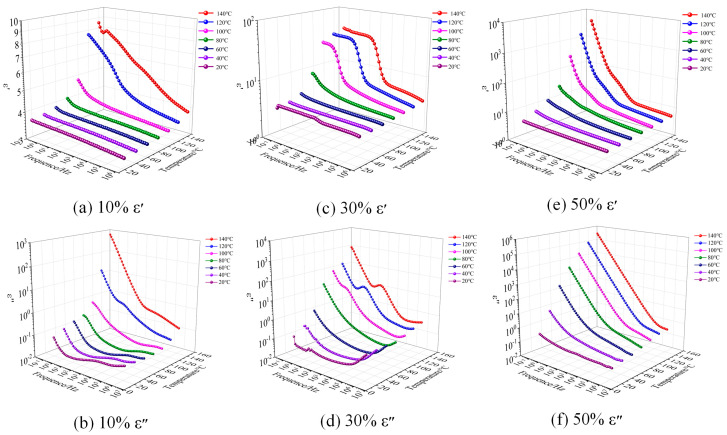
The frequency spectra of the real part ε′ and the imaginary part *ε*″ of the complex permittivity of EP with different PEGDGE contents at different temperatures.

**Figure 5 polymers-17-01405-f005:**
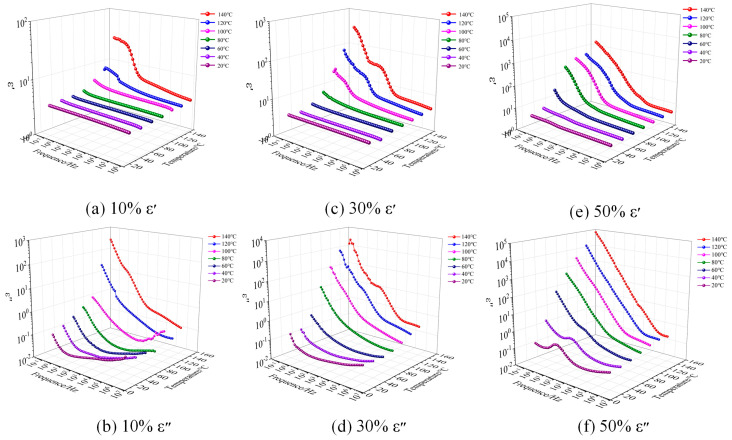
The frequency spectra of the real part ε′ and the imaginary part ε″ of the complex permittivity of EIP with different PEGDGE contents at different temperatures.

**Figure 6 polymers-17-01405-f006:**
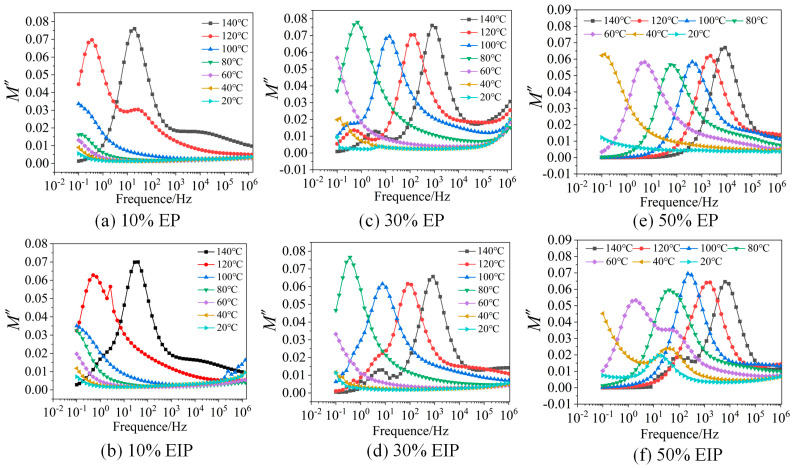
Frequency spectra of the imaginary part of complex electric modulus of EP composites with different PEGDGE contents at different temperatures.

**Figure 7 polymers-17-01405-f007:**
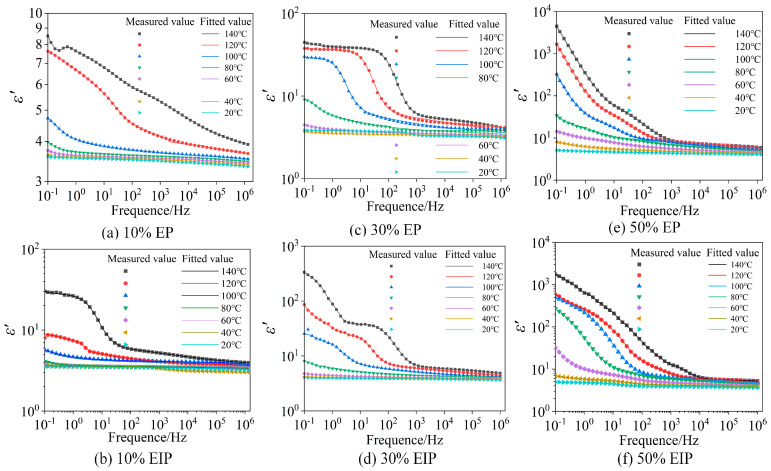
Fitting results of the real part of complex dielectric constant.

**Figure 8 polymers-17-01405-f008:**
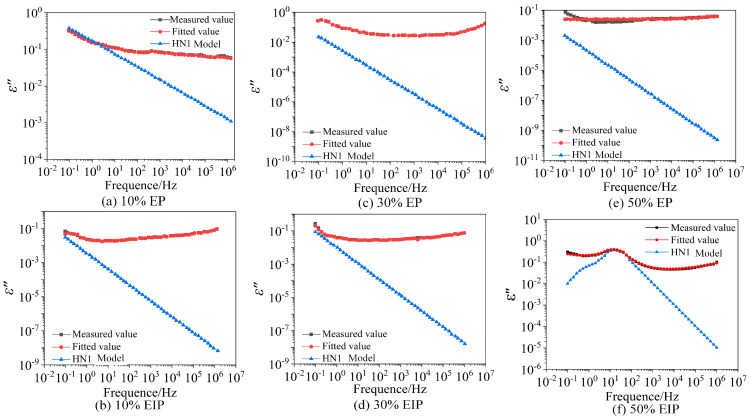
Fitting results of the imaginary part of complex dielectric constant at 20 °C.

**Figure 9 polymers-17-01405-f009:**
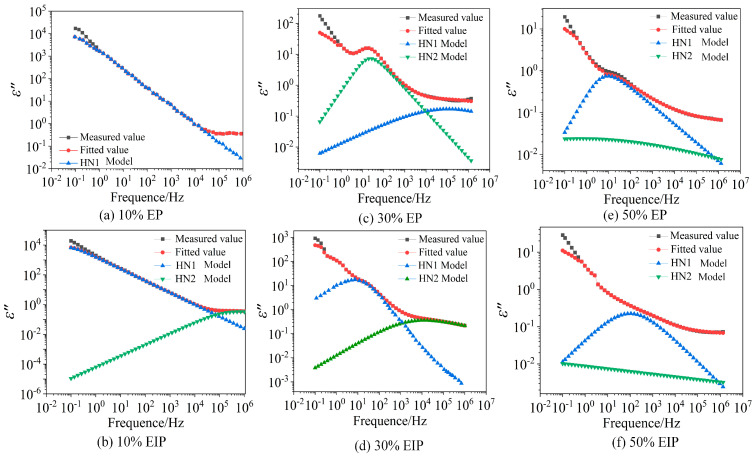
Fitting results of the imaginary part of complex dielectric constant at 120 °C.

**Figure 10 polymers-17-01405-f010:**
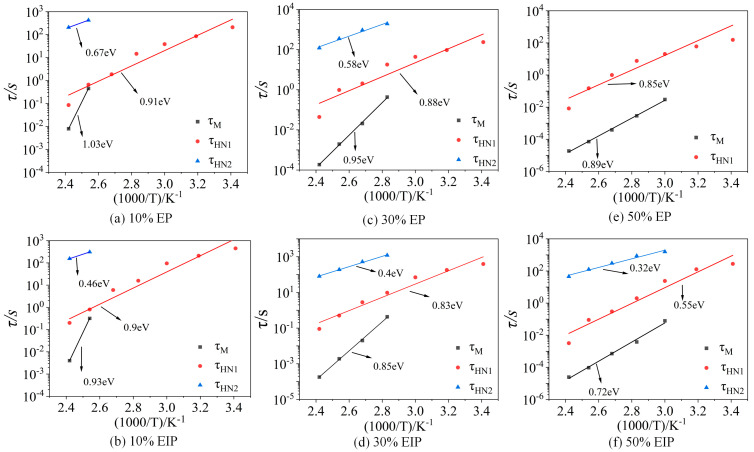
Variation in time constant of the dielectric relaxation process of EP composites with temperature.

**Figure 11 polymers-17-01405-f011:**
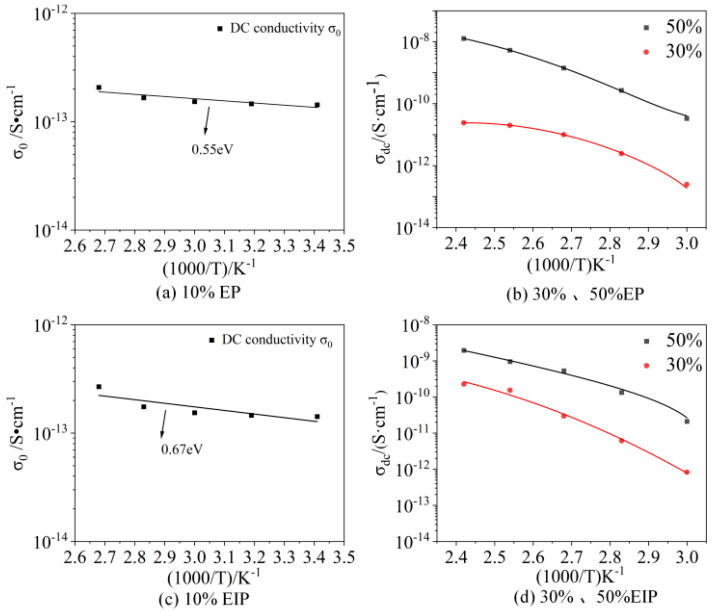
Variation in the DC conductivity of samples with temperature during the dielectric relaxation process.

**Table 1 polymers-17-01405-t001:** Mechanical properties of EP with different PEGDGE contents.

PEGDGE Content	Maximum Tensile Load/N	Tensile Strength (MPa)	Elongation at Break (%)
10%	2323	45.0	3.0
30%	1921	36.5	3.5
50%	1301	24.3	11.5

**Table 2 polymers-17-01405-t002:** Vogel temperatures T_0_ and glass transition temperatures Tg of PEGDGE/epoxy composition with different PEGDGE contents.

PEGDGE Content/Sample	Tg/K	T_0_/K
30%EP	332.95	382.78
50%EP	321.95	371.28
30%EIP	326.65	376.23
50%EIP	313.35	363.33

## Data Availability

The original contributions presented in this study are included in the article. Further inquiries can be directed to the corresponding author(s).

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
