# Peer review of "Quantitative Dissection of Relaxation Processes in Hybrid Epoxy Composites: Combining Dielectric Spectroscopy with Activation Energy Analysis"

_polymers, 2025, doi:10.3390/polym17101405_

Round 1
Reviewer 1 Report
Comments and Suggestions for Authors
The authors investigated the plasticizing effect using PEGGE in epoxy resins in order to study their effect in the relaxation process and Tg, which influences the dielectric properties. It is an interesting study; however, some aspects should be addressed before being considered accepted for this journal:
- In the introduction, please specify the meaning of abbreviations such as UHVDC in line 30.
- The requirement in terms of mechanical properties is recommended to understand if it could be used as a UHV converter transformer, at least to show the minimum required. If so, I would suggest including a quick test to corroborate in terms of mechanical capabilities.
- In a changing world toward a green environment, the authors should include why they chose EP with plasticizer (PEGGE) and whether these materials come from renewable resources and contribute to a better world.
- It is highly recommended to delve into how EIP preparation is made and make it clearer in sample preparation.
- Please indicate why the mixing ratios of curing agent and accelerator were selected. If necessary, please provide references.
- Please, revise in the whole document that every Tg is well written.
- Regarding test results, there is a lack of comparison between other papers published. The dielectric constant is as high as other developed materials?
- Please, provide the DSC graph. In addition to that, has it been proved that final resins are fully cured? It would be interesting to include a DSC and/or gel content to prove the crosslinking level.
- My recommendation, at least in the caption or figures 3, 4 and 5 is to include the Tg
- The dielectric loss factor with 3-D plots log (e’’), log (hz) and temperature, leads to observe better the relaxation, thus I recommend this graph at least in the supplementary material.
- Concerning the Arrhenius relationship, please include references in order to correlate the range of values for materials already tested. I also recommend converting it a kJ/mol. If not, please include references converting that value.
- I suggest including what the further studies are.
In general terms, it seems to be a potential paper, but more references should be included to corroborate each effect as well as comparation in order to show the potential of these materials.

Reviewer 2 Report
Comments and Suggestions for Authors
The authors present studies of epoxy materials with additives and study the effect on dielectric properties. However, in my view, there is insufficient information presented in this report to justify publication as is. A lists of comments is below, but two important points need to be improved. First, additional characterization and discussion is needed regarding the synthesis, structure, and morphology of the samples. Second, the fitting of dielectric data is questionable and needs to be further discussed and presented. In some plots, the "fits" capture features that the equations cannot (defects in the measurements) and in others, the additional HN function is not necessary. There are far too many parameters used to fit the data. Generally, the fitting needs to be discussed and presented more thoroughly including residual plots, goodness of fits, comparisons of different functions (one HN vs two), and a full list of final fit parameters.
List of comments that need to be addressed (in addition to above).
- Need to Define UHVDC and also re-define acronyms in the introduction (EP)
- There is no characterization of the epoxy to verify that the synthesis was successful and as intended. Similarly, there is no mechanical properties presented even though the point of these materials is toughening. Were the materials tougher?
-Please clearly describe the samples of EP vs EIP in terms of concentration of all additives (E51, curing agent, accelerator, PPGDE) I did not see definition of EP vs EIP formulations. Perhaps a table would be good.
- Please plot Tg vs concentration for. The sharp drop followed by the plateau, given only three data points, is not described sufficiently. Please site similar results in literature.
- Moving to the high frequency direction, line 169. What does this mean? Please reword.
- M’’ vs. tan delta in line 183. How are you defining these?
- What is meant by "destroys the cross linking structure" in line 209. Please discuss structure of the epoxy (with literature verification).
- What charges does the author mean? Impurities? Or were they deliberately added?
- Why are some epsilon plots log-log and others are semilog. It makes comparisons more difficult. All should be log-log.
- Figure six does not appear to be fits. I hope the authors are not trying to deceive the reader? Please show residuals for all curves. Also report a goodness of fit through R^2 value. In 6c, data from 120 and 140 C at low frequency show the “fit line” capturing defects in the data. This is also true for 6d at 100C and 120C
- The fitting in figure 7 is not appropriate. An HN relaxation cannot be observed in the data. And, for 7b, why does the HN1 model have a shoulder on the peak?
- The fitting in Figure 8 is not appropriate either. In 8d, how is the HN2 contribution larger than the fitted value? Also, the HN2 in Figure 8f cannot be used for analysis because the signal is too low and the peak is too broad – it is not capturing any features of the data and have four fit parameters.
- Please provide a table of all fit parameters used for each data set to verify results and ensure repeatability.
- If the HN1 is alpha relaxation, why is it Arrhenius near Tg?
- Consider normalizing plots to Tg to show appropriately how different the samples are. With a difference in ~60C in Tg between samples, the conductivity is expected to be different.
Comments on the Quality of English LanguageThere are typos in this draft, and explanations are in some parts confusing. But my primary concern is with the content.
Round 2
Reviewer 1 Report
Comments and Suggestions for Authors
According to the review, I would consider to publish this paper after incorporating two recommendations. A mistake is found in Fig 3, where caption repeat three times 10% EP. So please, revise. Regarding the mechanical properties, it seems that only EP has been done but, what about EIP?
